

**Towards a monitoring system of temperature extremes in Europe**
Christophe Lavaysse[1], Carmelo Cammalleri[1], Alessandro Dosio[1], Gerard van der Schrier[2], Andrea Toreti[1] and
Jürgen Vogt[1]
Affiliations:
1- European Commission, Joint Research Centre (JRC), Ispra, Italy
2- Royal Netherlands Meteorological Institute (KNMI), De Bilt, The Netherlands
Abstract
Extreme temperature anomalies such as heat and cold waves may have strong impacts on human activities and
health. The heat waves in Western Europe in 2003 and in Russia in 2010, or the cold wave in South-Eastern
Europe in 2012, generated a considerable amount of economic loss and resulted in the death of several
thousands of people. Providing an operational system to monitor extreme temperature anomalies in Europe is
thus of prime importance to help decision makers and emergency services which are responsive to an
unfolding extreme event.
In this study, the development and the validation of a monitoring system of extreme temperature anomalies
are presented. The first part of the study describes the methodology based on the persistence of events
exceeding a percentile threshold. The method is applied to three different observational datasets, in order to
assess the robustness and highlighting uncertainties in the observations. The climatology of extreme events
from the last 21 years is then analysed to highlight the spatial and temporal variability of the hazard and
discrepancies amongst the observational datasets are discussed. In the last part of the study, the products
derived from this study are presented and discussed regarding previous studies. The results highlight the
accuracy of the developed index and the statistical robustness of the distribution used to calculate the return
periods.

Key words: Monitoring heat waves, EOBS, ERAI, Europe
Corresponding author: Christophe Lavaysse, European Commission, Joint Research Centre (JRC), Directorate
for Space, Security and Migration, Disaster Risk Management Unit, Via E. Fermi 2749, I-21027 Ispra (VA),
Italy
Christophe.lavaysse@ec.europa.eu





## 1 Introduction

Extreme temperature anomalies have strong impacts on human health and activities. The heat waves that occurred over Western Europe in August 2003 caused about 70,000 deaths across twelve countries (Robine et al. 2008). The heat wave in Russia during the summer 2010, considered as the strongest in the last 30 years (Barriopedro et al. 2011, Russo et al 2015), caused more than 55,000 victims and 500 billion euro of damage. In February 2012 a cold wave over Central and Eastern Europe generated more than 700 million euro of damage, and 825 deaths were reported (de'Donato et al., 2013). Monitoring and cataloguing these events are crucial in order to place an event in the historic perspective and in order to assess the potential impacts on human health and activities by combining the information with data from other catalogues (such as EM-DAT, http://www.emdat.be). A catalogue would also be appropriate to analyse the spatial and temporal evolutions of the hazard related to temperature anomalies, and, finally in the future, to calibrate and validate an operational forecasting system in terms of these extreme events. This product will be implemented in the operational monitoring system of the European Drought Observatory (EDO, http://edo.jrc.ec.europa.eu).

From the human health point of view, a heat (cold) wave can be considered as a period with sustained temperature anomalies resulting in one of a number of health outcomes, including mortality, morbidity and emergency service call-out (Kovats et al., 2006). Wave intensity and duration, but also time of the year, are important determinants of the impact on health (Montero et al., 2012; Rocklov et al., 2012). While most studies focus on daytime conditions only, there is emerging evidences that nocturnal conditions can also play an important role in generating heat-related health effects, a result of the cumulative build-up of the heat load with little respite during the night (Rooney et al., 1998).

In the literature, some indicators have been developed to describe the complex conditions of heat exchange between the human body and its thermal environment. For warm conditions, indices usually consist of combinations of dry-bulb temperature and different measures for humidity or wind speed, such as: the humidex (Smoyer-Tomic et al. 2003), the net effective temperature (Li and Chan, 2000), the wet-bulb globe temperature (Budd, 2009), the heat index (Steadman, 1979) or the apparent temperature (Steadman, 1984). More generally, efforts have been made to harmonize the large number of indices developed. For example, the Universal Thermal Climate Index (UCTI, www.utci.org) has been proposed to assess heat and cold waves. The main inconvenience of most of these indices is technical, i.e., the humidity when the daily maximum or daily minimum temperature (hereafter Tmax and Tmin) occur is not necessarily known. In addition, the simulate values of wind speed and humidity provided by numerical weather models are generally less accurate than the 2m temperature in the reanalysis and observational datasets. The WMO Expert Team on Climate Change Detection and Indices (ETCCDI) proposed the Warm Spell Duration Index (WSDI) as standard measurement of heat and cold waves which is calculated using a percentile-based threshold. Russo et al. (2015) proposed a version of this method that provides the amplitude (or intensity) of a heat wave based on the maximum temperature and the interquartile range of yearly maximum temperature of the past period. This method is powerful to compare the heatwaves at climatological scale over the world and their trends with a local standardization. Nevertheless, this method is not suitable for monitoring heat waves because it focuses





on the most extreme events (the thresholds are defined according to the yearly maximums), and it does not
take into account the Tmin that has a strong human impact according to WMO (2015).
In this study we propose an operational system to monitor heat and cold waves based on an adapted index
inspired by the previous studies. In section 2, data and methods are presented and the uncertainties related to
the observations are assessed. Then, the climatology in term of occurrence, intensity and duration of the waves
are presented in section 3. This represents the baseline of the monitoring system that will become operational
and embedded in the EDO system. Finally, concluding remarks are provided in section 4.

**2      Data and tools**
**2.1      Datasets**
In this study we use daily Tmax and Tmin from three different datasets. The first one is based on the 2m
temperature datasets provided by the European National Weather Services, which, in turn, is used as an input
for the LisFlood hydrological model (De Roo et al., 2000). The observations are gridded onto a regular lat/lon
grid of one square degree. The use of gridded observation data is motivated i) to focus on large scales heat/cold
waves and ii) to compare the station data with reanalysis. This LisFlood product will be eventually used in the
operational system for the monitoring of extreme temperature waves. To validate the results, a comparison
with two other sets of data is performed: the ERA-Interim reanalysis (ERAI, Dee et al., 2011) and the
EOBS/ECAD dataset Version 14 (Haylock et al., 2008, van den Besselaar et al., 2011), both regridded to the
same one square degree resolution. Note that, according to ECMWF, ERAI datasets are released with a delay
of two months for quality assurance; as a consequence this dataset cannot be used for operational monitoring
purpose. The same problem occurs for the EOBS datasets.
The definition of Tmax and Tmin in the three datasets can differ from the definition of WMO (van den
Basselaar et al. 2012). In LisFlood, the Tmin assigned to the day d is defined as the minimum temperature
value that occurred from 1800Local Time (LT) of the day before (d-1) to 0600LT of the day d. For EOBS,
Tmin is defined as the 24-hour daily minimum. Similarly, Tmax of the day d is the maximum temperature
recorded from 0600LT to 1800LT of the day d for LisFlood data and the 24-hour daily maximum for EOBS.
In ERAI, Tmin (Tmax) of day d is the lowest (highest) value of temperatures recorded at 0000LT, 0600LT,
1200LT or 1800LT of day d. The starting years of the period covered by the datasets are also different (1950
for EOBS, 1979 for ERAI and 1990 for LisFlood). In order to be consistent and in a view of the future use for
the reforecast period of the ECMWF ENS forecast model, the period from 1995 to 2015 (21 years) is used for
all the datasets. Note that most of the results obtained in this study have been compared to a longer period
(starting from 1990) providing very similar results.
**2.2      Metric of extreme temperature anomalies**
Following the WMO definition, there are many different ways to measure a heat wave (Perkins et al., 2013).
The objective of this study is not to create a new index, but to provide an operational system based on an





adapted method proposed in the literature. This system is inspired by the studies of Russo et al., (2014) and
WMO (2015). First, daily Tmin and Tmax are transformed into quantiles based on the climatological (21
years) calendar percentiles of each variable. To highlight the events with the most potential human impacts,
the year is cut in two periods: The extended summer period, when heat waves could have more impacts (6
hottest month over Europe, from April to September), and the extended winter period to focus on the cold
waves (from October to March). Note that also during the summer (winter) period, cold (heat) waves may
occur but they are not considered here. The independent calculation of the daily quantiles of observed Tmin
and Tmax is done by applying a leave-one-out method to avoid inhomogeneities (Zhang et al. 2005). The year
studied is removed from the climatology. The data without this year is exploited to perform the observed
cumulative distribution function (CDF). To remove artefacts due to the relative small sampling (21 years), a
window of 11 days centred on the day studied is exploited. The daily temperatures are transformed into
quantile by this procedure to create two daily temperature quantiles from 1995 to 2015, derived from the CDF
of Tmin and Tmax independently.
The main difference with the previous studies is the use of both Tmax and Tmin, rather than Tmax only or the
daily mean temperature. Then a hot day is defined when simultaneously the daily quantiles of Tmax and Tmin
are above quantile 0.9 during the extended summer (from April to September). The same definition is applied
for cold days when the two quantiles are lower than quantile 0.1 from October to March. The occurrences are
strongly influenced by these thresholds. As this study aims at quantifying the intensity of waves regarding the
climatology and at assessing with robust scores the forecast of these events, it is not possible to focus only on
the most extreme cases. So these thresholds (quantiles 0.9 and 0.1) are chosen as compromise between the
need to have a minimum number of events and the definition of extremes. They are also used in a large number
of other studies (WMO, 2015, Hirschi et al., 2011). Note that in order to discuss the sensitivity of using the
intersection of Tmin and Tmax rather than one temperature value per day, the same methodology has also
been applied using only Tmin and only Tmax to determined hot and cold days.
Heat and cold waves are associated with a persistence of hot or cold days. Based on the literature (Gasparrini
and Armstrong, 2011, Kuglitsch et al, 2010), as well as on the recommendation of WMO (2015) for health
impacts, we define a heat (cold) wave as an event of at least 3 consecutive hot (cold) days (i.e. when
simultaneously Tmin and Tmax exceed the quantile thresholds). A pool is also introduced when two events
are separated by less than one day. Note that periods in between two waves are not taken into account in the
wave duration and in the wave intensity. Fig. 1 illustrates the method used to detect heat waves in this study.
The European mean distribution of these cases are presented in Table 1 and 2 using the LisFlood dataset, but
the results are very similar with the two others datasets (not shown). The table demonstrate the impact of using
the intersection of Tmin and Tmax above (below) the thresholds. With respect to heat waves (Table 1), for
example, about 150 out of 376 days, i.e. 40% of the Tmin above the thresholds occurred simultaneously (i.e.
the same day) with Tmax above the threshold (Table 1, first column). Even if the differences are not large,
there is another result with a significant more persistency of Tmax than Tmin. For instance, using Tmax only,
70% of the hot days (269 out of the 382) are detected as being part of a heat wave, whereas using Tmin only,





the ratio is about 60% (i.e. 226 out of 376). Using both Tmax and Tmin, on average 81.3 days (54% of the hot
days) are detected as being part of a heat wave (Table 1, second column). Finally, the mean occurrences of
heat waves are indicated in the last column. The use of the two temperatures tends to reduce drastically the
number of events (from 44 or 51 to 16.9 on average during the period) but also their durations (5.11 or 5.3
days to 4.8). The continental regions appear less affected by this reduction than coastal regions (not shown).
In analogy, Table 2 shows the same data for the case of cold waves.
Once a wave is detected, two main characteristics are recorded: the duration (in days) and the intensity. To
take into account different characteristics and to assess the sensitivity of the methods, the latter is calculated
by three different methods. The first one is based on the sum of the quantiles above (or under) the threshold
during the detected wave.

$$I1(n) = \sum_{i=1}^{N} \beta \frac{[Qtx_{i,w} - Thres + Qtn_{i,w} - Thres]}{2} \begin{cases} \beta = 1 \; for \; Heat \; waves \\ \beta = -1 \; for \; cold \; waves \end{cases}$$


Where I1 is the intensity of the wave having duration equal to N days, Qtn and Qtx the daily quantile of Tmin
and Tmax and Thres, the quantile thresholds (i.e. 0.9 and 0.1 for heat and cold days respectively). The purpose
of dividing this intensity by 2 is to create an intensity comparable to the intensities calculated with Tmin and
Tmax only. The second method is similar to the first but the quantile differences are replaced by the
temperature anomalies with respect to the climatological daily thresholds. This method is defined as follows:

$$I2(n) = \sum_{i=1}^{N} \beta \frac{[Tx_{i,w} - Q_{Tx} + Tn_{i,w} - Q_{Tn}]}{2} \begin{cases} \beta = 1 \; for \; Heat \; waves \\ \beta = -1 \; for \; cold \; waves \end{cases}$$

Where $Q_{Tx}$ and $Q_{Tn}$ represent the calendar daily thresholds of Tmin and Tmax, i.e. the temperatures for the
quantiles 0.9 (0.1) for the heat (cold respectively) waves. This method allows quantifying intensities regarding
the seasonal cycle and reflects an anomaly but not necessarily extreme values of absolute temperatures. This
calculation is motivated for agricultural applications, where the yield crops can be sensitive to these large
anomalies during the transitional seasons (Porter and Semenov, 2005). The last method is also based on
temperature anomalies but regarding a constant threshold.

$$I3(n) = \sum_{i=1}^{N} \beta * [\frac{[Tx_{i,w} - Tx_{med(Q_{Tx})}]}{2 * \sigma_{Tx}} + \frac{[Tn_{i,w} - Tn_{med(Q_{Tn})}]}{2 * \sigma_{Tn}}] \begin{cases} \beta = 1 \; for \; Heat \; waves \\ \beta = -1 \; for \; cold \; waves \end{cases}$$






Where $Tx_{med(Q_{Tx})}$ and $Tn_{med(Q_{Tn})}$ represent the constant temperature of the median of all the calendar daily
quantiles of 0.9 (heat waves) and 0.1 (cold waves) of Tmax and Tmin. $\sigma_{Tx}$ and $\sigma_{Tn}$ represent the
climatological yearly variance of Tmax and Tmin. This method is intended to increase the intensities of a
heat or cold waves that occur close to the maximum or minimum of the seasonal cycle. Based on this
calculation, the strongest intensities are generally associated with the warmest or coldest absolute
temperatures. The division by the variance of the seasonal cycle is justified to reduce the intensity of the waves
that occurred over region with strong seasonal cycle, where the variability of temperature is well known to be
variable. The latter method is conceptually close to the one proposed by Russo et al. (2015) and, due to its
sensitivity to the absolute temperatures, could be more suitable to assess the potential impacts on human
health. Fig.1 illustrates the heat wave detection and the calculation of the two last methodologies. The different
intensities provided by these three methods, which use the same detection method, are discussed in the results
section.

**3      Results**
**3.1      Comparison of the datasets**
In order to compare the observations and quantify the uncertainties of the results, different datasets, provided
by observations and reanalysis, are used. First, the temporal correlations between different pairs of the daily
quantiles are shown in Fig. 2. We notice that the correlation of the quantiles of Tmin and Tmax from ERAI,
EOBS and LisFlood datasets are quite in agreement (the spatial mean correlation is about 0.89). Note that as
the quantiles are used, the seasonal cycle is removed, showing the quality of this agreement. The scores are
generally better for Tmax than Tmin. This could be explained by the larger spatial homogeneity of Tmax than
Tmin and the differences in the Tmin definition amongst National Weather Services. Indeed, over certain
countries, Tmin is measured during night time between 1800LT and 0600LT the following day, elsewhere
from 0000LT to 2400LT or from 0600LT day d to 0600LT day d+1, which could result in a delay of one day.
In the EOBS data description, and in van den Besselaar et al. (2011), this point and the uncertainties associated
are deeply analysed. Due to the coarser resolution and only 4 recorded values per day to calculate Tmin and
Tmax, ERAI is associated with a hot bias of Tmin and a cold bias of Tmax in relation to both LisFlood and
EOBS datasets (not shown). The yearly Mean Absolute Errors of Tmin and Tmax (MAE, Fig. 3, very close
to the Root Mean Square Differences) remains, however, relatively low (<1.5 deg.) except at the borders of
the domain, confirming the good agreement especially between EOBS and ERAI. Note that the LisFlood
dataset is slightly less correlated to the others over Scandinavia, Germany and on the North-easternmost part
of the domain probably due to the definition of Tmin and Tmax for each country, delay in the GTS
communications and the density of the stations (the E-OBS network over Germany and Scandinavia is quite
dense).





### 3.2  Climatology

#### 3.2.1  Occurrence variabilities of the waves

The total occurrences of heat and cold waves during the 21 years are calculated using the definition presented in section 2. This is performed independently for the three datasets to provide information on the robustness of the results. As shown in Table 1 and 2 cold waves are more frequent than heat waves for the three datasets especially in the eastern part of Europe (Fig. 4 and 5, first row). The same figures using independently only Tmin or Tmax to detect both heat and cold waves reveal more homogeneous spatial patterns and quite the same number of occurrence between them, but about 50 to 60% more than the intersection of Tmin and Tmax (Fig. 4 and 5 second and third row). Only the detection of the heat waves using Tmin only generate less events. These results highlight two main characteristics: 1) the lower persistency of Tmin with strong positive anomalies could partially explain the difference between the occurrence of heat and cold waves; 2) the increase of the occurrence in the continental regions is mainly explained by an increase of the simultaneous anomalies in Tmin and Tmax rather than an increase of occurrence of persistency. These two characteristics may be explained by the synoptical situations during cold waves and the fact that there are more frequent meteorological blocking conditions in winter than in summer (Tibaldi et al. 1994, Doblas-Reyes et al, 2002). Several recent studies (Tomczyk and Bednorz 2016, Sousa et al. 2017) emphasized the important role of persistent and intense blocking and associated anticyclones in producing heat or cold waves. The origins of the extreme blocking situations are still not well understood and could be related to the development of a large-scale Rossby train (Trenberth and Fasullo 2012). Finally the study of Schubert et al. (2014), who identified Western Russia as the leading mode of surface temperature and precipitation covariability could highlight the potential feedback of the soil moisture in enhancing the intensities of the heat waves over this region (Fisher et al. 2007, Mueller and Seviratne 2013, Miralles et al. 2014, Whan et al. 2015).

The main difference between the datasets is the higher occurrence of both heat and cold waves for ERAI than the other datasets. This could be an effect of the coarser resolution in time and space of the reanalysis compared to the ground observations that tends to smooth the temporal evolutions of the temperature anomalies and so of the quantiles. Due to that lower temporal variability, the chance to get long term anomalies is increased when using ERAI as compared to the other datasets.

As the total number of occurrences is the sum of all individual waves, the distribution of the wave durations is needed to complete the picture. Fig. 6 displays the spatial variability of the last quartile of the wave durations recorded for each grid point. It appears that the difference between the durations of heat and cold waves between the three different datasets is much lower than the difference of occurrence discussed previously (Fig. 4 and 5). It is also interesting to note that, especially for cold waves, the regions where the waves are the most frequent are not the same where they are the most persistent. Finally, it is remarkable to record many of the longest durations of the cold waves along the coasts of the North Sea and the Baltic Sea. Indeed, the climate along the coasts is generally more variable than in the continental regions, and so the waves are supposed to be shorter. According to the same calculations using only Tmin and Tmax (not shown), the spatial

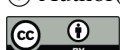



heterogeneity of the cold wave durations is much larger when Tmax is used than Tmin and we observe a strong
increase of the wave durations with Tmax over northern Germany, Denmark, northern Poland, the Baltic Sea
and southern Scandinavia. That highlights the persistency of negative anomalies of Tmax over these regions,
which could increase the chance to get longer durations with the intersection method and could explain the
results in Fig. 6.
**3.2.2    Intensity of the heat and cold waves**
The climatology of the intensities is important but very sensitive to the definitions applied in order to provide
a baseline and to calibrate the waves monitored, it is, therefore, important to analyse and justify the method
applied. The three methods, I1, I2 and I3 (using the quantiles, the temperature anomalies and the constant
threshold of temperature, see Sect. 2.2), are compared during heat and cold waves in Fig. 7. The distributions
of each scatter plot indicate the relationships by pairs in between the three methods for all the events, and the
colours indicate the corresponding durations of the events. Note that this figure is using LisFlood, but the same
results are obtained for the other datasets. These panels show the strong dependency of the intensities derived
from the quantiles and the durations (colour distribution well vertically distribute in Fig. 7b and horizontally
in Fig.7c and 7e). This is especially true for the cold waves (correlations in between duration and I1 larger
than 0.95). These high correlations highlight the redundancy in the information with the wave durations.
Moreover, I1 is also climatologically bounded by the values recorded during the past period. For these reasons
the use of the quantiles appears not suitable to assess the heat and cold wave intensities. The methods derived
from the temperature anomaly (I2) and constant threshold (I3) are chosen. Indeed, the correlations in between
the wave durations and I2 and with I3 are much lower and not significant (on average 0.72 and 0.59), showing
the potential additional information provided. Moreover, these values are not bounded by the historical values
and so they will be able to better distinguish the most severe cases. According to the scatter plots in Fig. 7d
(for the heat waves) and Fig. 7f (for the cold waves), these methods appear quite independent at European
scale. Nevertheless the analyses of the correlations in between the methods a grip point level reveal a large
spatial variability (not shown). For instance, the correlations of I2 and I3 go up to 0.95 over France and
Western Russia, explained by heat (cold) waves occurred during the warmest (coldest) months, and go down
to 0.5 over Central and Northern Europe.
Except for the strongest events, there is an overall good agreement of the datasets to represent the probability
distribution functions (PDF) of the intensities of heat and cold waves. For instance, Fig. 8 display the
distribution of intensities defined by the method of the temperature anomalies (I2) and show no significant
difference for intensities lower than 60. This figure also confirms our finding of the higher occurrence of cold
waves than heat waves especially in the intensities larger than 25. In the tails of the distribution (especially
for the heat waves larger than 90), the differences are associated with a very low number of cases. The spatial
variability of these intensities I2 for the last 21 years was assessed by the strongest cold and heat waves
recorded over each grid point (Fig. 9). From this figure, the two strongest heat waves that occurred in Europe
can be clearly identified, namely the one that occurred in Russia in 2010 and the one in France in 2003. For
these two events, the intensities are slightly stronger using ERAI with both a longer duration and daily





intensities (not shown). For the cold waves, the intensities are stronger than the heat waves. The most intense
events occurred over the continental regions (Central Europe and South of Russia). The three datasets are well
in agreement for the intensities and the spatial variabilities. It is interesting to highlight that these intensities
are not well correlated to the occurrence, i.e., a region with more cases does not necessarily record the most
extreme events (Fig. 4 and 5). The relative short period of study (21 years) can generate some artefacts over
regions that recorded extraordinary events (especially true for Russia).
To assess the uncertainties relative to the methodology used, Fig. 10 display the same distributions but for
intensities calculated using constant thresholds (I3). Even if the scales are different, the spatial distribution of
I2 and I3 for the strongest heat waves are quite similar. The patterns are strongly influenced by the two heat
waves in 2003 and 2010. In opposite, the distribution of the strongest cold waves changes drastically. Whereas
the intensities over Russia are reduced, we note a relative increase of the intensities in Western Europe,
especially in North Germany, the Netherlands, and in Central Europe. As discussed previously, this
modification is explained by events that occurred more during the transitional months (intense I2 but not I3)
or close to the maximum (or minimum) seasonal temperature (intense I3). The spatial distribution is also
changed due to the normalisation according to the amplitude of the seasonal cycle, which is larger in
continental regions (not shown). Even if the results display significant differences according to the methods
and the regions, it is important to note that the three datasets are still well in agreement.

### 3.3    Return periods

As the purpose of this study is to provide a methodology that is useable for a monitoring system that must be
robust and understandable for users and decision makers, the information will be provided in terms of return
periods. This product will quantify, at monthly time scale, the intensity of the cold or heat waves that have
occurred. To build this indicator, all the days defined as cold or heat waves are summed for different
accumulation periods (from monthly to seasonally, see Table 3). Monthly values characterize either one
specific event as defined previously or several consecutives cases. As indicated by WMO (2015), intense or
repetitive extreme waves may have strong impacts on human health and so should be assessed. Once these
monthly values are calculated for each grid point, the return period can be estimated. Problem when dealing
with extremes are linked to erroneous extreme values and the sampling. To partially address these issues, we
have compared different datasets and different theoretical distributions have been fitted and tested on the
empirical distribution of the summed up intensities. This is done at grid point and at region level. In the
literature, different distributions have been tested such as the Gamma (Meehl et al. 2000) or the Weibull
distribution (Cueto et al 2010). According to the Pearson goodness-of-fit statistic, and the deviance statistic
on the entire distribution, the Gamma distribution is the most robust (not shown). By using the theoretical
distribution, the return periods can be extrapolated to durations longer than the duration of the initial time
series (21 years). Despite the non-stationarity of the climate, this relative short climate period reduces the
trend and allows the hypothesis of stationarity. Once the parameters of the Gamma distribution are estimated
for monthly, bimonthly and seasonal time scales (see table 3), return periods are calculated for both the cold



and heat waves. In that section, the results are produced using LisFlood dataset, which has been validated in
the previous section, but similar results are obtained with the two other datasets.

The boxplots (in Fig. 11) indicate the relationships between intensities and return periods over each grid point
in Europe. According to size of the inter quartiles, it appears the large spatial variabilities over the domain.
For instance, heat waves with intensities of 20 (10) using I2 (I3) have inter quartiles of return period that span
from 7 to 50 years (25 to 125 years respectively). Due to the more frequent intense cold waves, the return
periods are smaller than for the heat waves for similar intensities. The use of other datasets provide similar
results. Nevertheless, according to the previous findings, ERAI has less spatial variability (lower spread of the
boxes), and lower return periods due to the larger wave intensities recorded (not shown).

The spatial variabilities are then analysed in more detail with a region classification. This classification is a
simplification of the one reported in the EEA report (2016) that takes into account the climatology of the
regions (Continental, Mediterranean, Oceanic, Scandinavian, small panels in Fig. 11). Over these regions, the
calculation of return periods are assessed and compared (coloured dots in Fig. 11). Even if the results for the
two intensities (left and right panels) cannot be compared directly because the calculations are different, it is
interesting to compare the order of the regions according to the return periods. For heat waves, the British
Islands are remarkable by using the two intensities. The few intense heat waves recorded generate return
periods in the outliers of the box distribution in Europe. In opposite the Russian region records the lowest
return periods for similar intensities using I2 showing the large hazard of these heat waves in this region.
Nevertheless, the use of the I3 calculation (more sensitive to waves that occurred during the hearth of the
season) shows a different distribution with more cases over Central Europe for return periods lower than 5
years (in yellow) and the North-West European region (red) for the most intense heat waves. For the cold
waves, the British Islands and the Mediterranean regions are the least affected in the two intensity calculations,
whereas the continental parts of Europe (Russia and Central Europe) are associated with more regular intense
cold waves.
Based on these calculations, the monthly intensities are transformed into return periods, resulting in more
comprehensible information for users. In Fig. 12, the intensities of the heat and cold waves with a return period
of 10 years are plotted using I2 and I3. These values are sensitive to the distribution (number and intensities)
of the waves recorded during the 21 years analysed. That is why we observe the increase of intensity over
western Russia in Fig. 12 (left panels) where the waves are more frequent (Fig. 4 and 5) and more intense
(Fig. 9). The same results with I3 show a different behaviour Fig. 12 (right panels), mainly due to the change
of the most intense waves recorded and plotted Fig. 10. The potential impacts of these heat and cold waves
will be calculated regarding the absolute intensities and the return periods. But we could expect that identical
intensities of waves that occurred over two different regions and so with two different return periods may have
different impacts. For example, the absolute value of the heat wave intensity recorded in August 2003 over





France using I3 does not give extreme value regarding the intensities recorded in continental regions.
Nevertheless, the equivalent return value over France is larger than 50 years (not shown), in agreement with
Barriopedro et al. (2011) and Trigo et al. (2005), which suggest the potential strong risk associated.
Given the short period used in this study, the return periods cannot be as accurate as those ones reported in
previous studies, nevertheless, the information allows identifying the most extreme situations. The same
information is also available for the 2-month and seasonal time scales (not shown). The developed
methodology will allow to provide a robust and understandable indicator that is standardized by the local
climatology.

## 4    Summary and Conclusions

In this paper, we propose to monitor heat and cold waves by using a method based on the persistency of the
exceedance of quantiles of daily minimum and maximum temperatures at grid point level. In the first step, the
methods to detect and quantify the intensities of heat and cold waves were assessed. Tests were performed
with three methods: using the sum of the quantiles, the sum of the temperature anomalies regarding thresholds
depending the calendar days and the sum of the temperature anomalies regarding a constant threshold. The
sensibility tests on the influence of Tmin, Tmax and the intersection of both demonstrated the large influence
of using the two daily temperatures in decreasing drastically their frequencies. Finally three datasets were
compared, two derived from station data (LisFlood and EOBS) and one from reanalysis data (ERAI). The two
observational datasets presented only minor differences in heat and cold waves occurrences and intensities.
This is probably due to the good agreement in between the two datasets for Tmin and Tmax. Using ERAI
some differences appeared mainly due to the coarser resolution of the original grid and the use of only 4 values
per day to define Tmin and Tmax. In this case, the persistency and the spatial correlation were increased,
generating less spatial distinction and more intense waves than using the first two datasets. However, the main
results are in overall agreement for all three datasets and show an increased hazard for heat and cold waves in
the continental part of Europe. The data are also in agreement when transforming the intensities into return
periods. These relationships will be used operationally in the EDO website to provide robust and
comprehensible information for decision makers and users.
In perspective, these datasets and results should be compared to the results derived from forecast products in
view of providing a comprehensive and seamless tool for monitoring and forecasting heat and cold waves in
Europe.

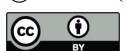



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

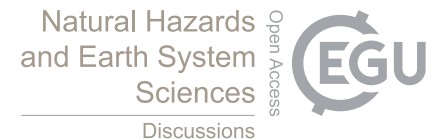

|         | HOT DAYS      | DAYS IN HW    | NUMBER OF HW  |
|---------|---------------|---------------|---------------|
| $T_{MIN}$ | 376 (17.9)  | 226 (31.8)    | 44.2 (5.1)    |
| $T_{MAX}$ | 382 (10.7)  | 269 (31.0)    | 51 (4.9)      |
| $T_{INT}$ | 150 (36.3)  | 81.3 (33.9)   | 16.9 (6.1)    |

Table 1 Spatial mean (and standard deviation in brackets) of number of days detected as hot days (larger than
quantile 0.9, first column), over the entire period of analysis, days detected during heat waves (HW, with
persistency longer than 3 days, second column) and total number of HW (third column) using only Tmin (first
row), only Tmax (second row) and the intersection of the two variables ($T_{int}$, third row).


|         | COLD DAYS     | DAYS IN CW    | NUMBER OF CW  |
|---------|---------------|---------------|---------------|
| $T_{MIN}$ | 380 (20.8)  | 272 (30.5)    | 50 (5.3)      |
| $T_{MAX}$ | 380 (14.8)  | 282 (27.4)    | 50.3 (4.3)    |
| $T_{INT}$ | 196 (48.2)  | 128 (42.7)    | 25.2 (7.6)    |


Table 2 Same as Table 1 for the cold days and cold waves (CW).


| Months   | JAN  | FEB  | MAR     | APR  | MAY  | JUN  | JUL  | AUG  | SEP     | OCT  | NOV  | DEC  |
|----------|------|------|---------|------|------|------|------|------|---------|------|------|------|
| Type     | Cold | Cold | Cold    | Heat | Heat | Heat | Heat | Heat | Heat    | Cold | Cold | Cold |
| Duration | 1, 2 | 1, 2 | 1, 2, S | 1    | 1, 2 | 1, 2 | 1, 2 | 1, 2 | 1, 2, S | 1    | 1, 2 | 1, 2 |


Table 3 Type of wave calculated operationally at the end of every Month. The period in which the intensities
are calculated are indicated in the last row (1 for 1 month, 2 for 2 months or S for Season, i.e. 6 months).









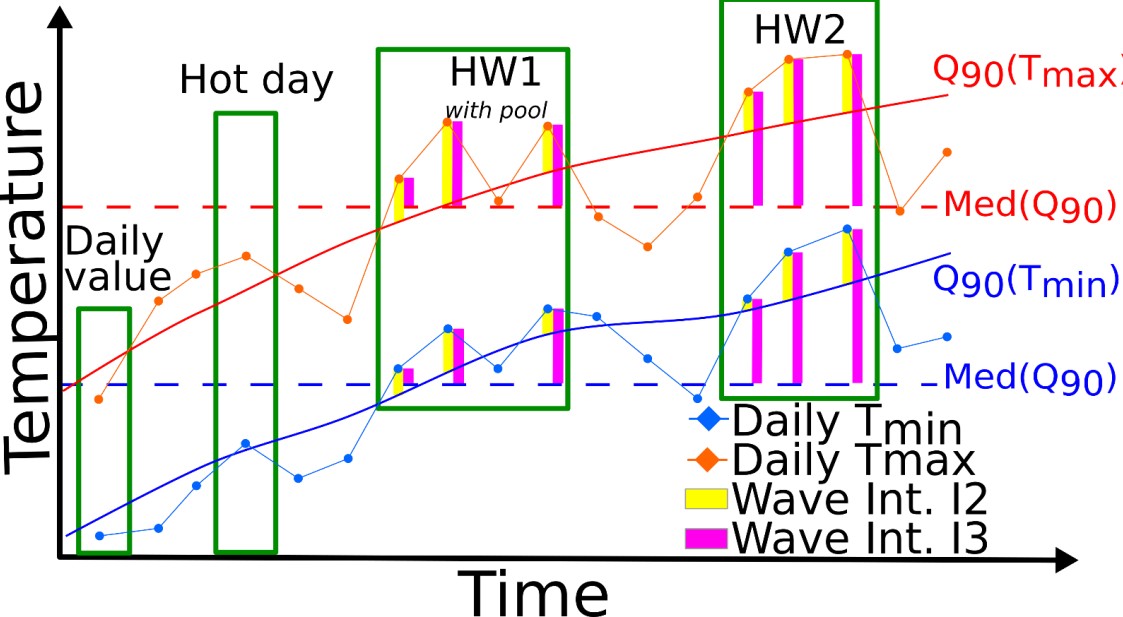


Figure 1 Schema of the detection method and the calculation of the intensities of heat waves, based on
temperature anomalies of a calendar day threshold: Q90 of both Tmax and Tmin (I2 calculation), or based
on the constant climatological threshold defined by the median of the daily quantiles: Med(Q90) of both
Tmax and Tmin (I3 calculation).





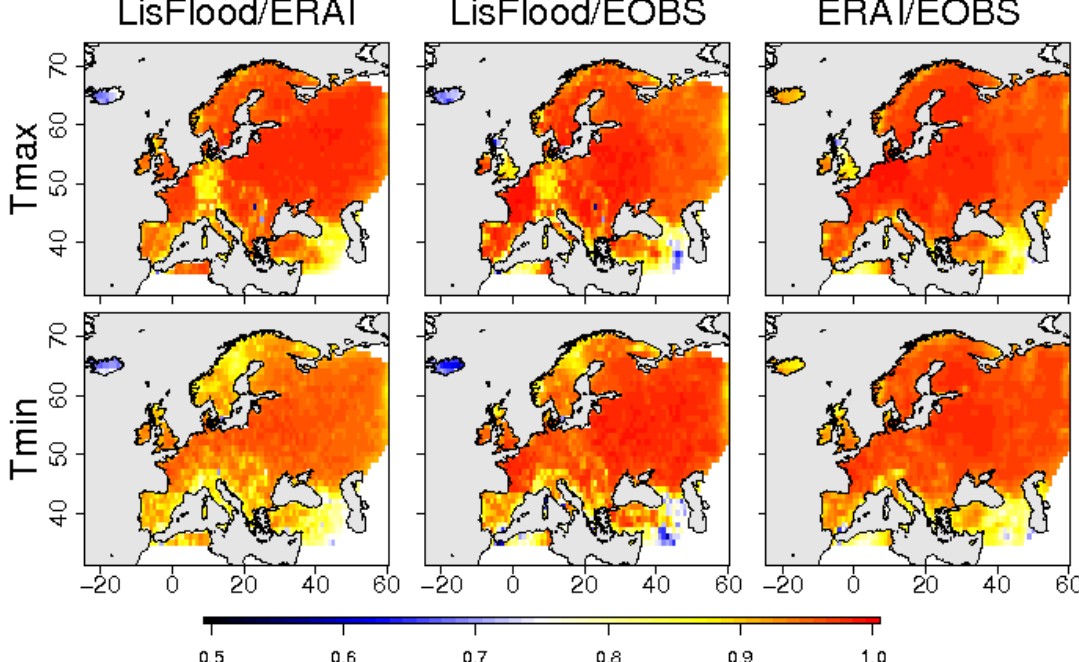


Figure 2 Temporal correlation of the temperature quantiles of Tmin (first row), and Tmax (second row)

provided by ERAI, EOBS and LisFlood datasets from 1995 to 2015. The datasets compared are indicated on

the top of each column.






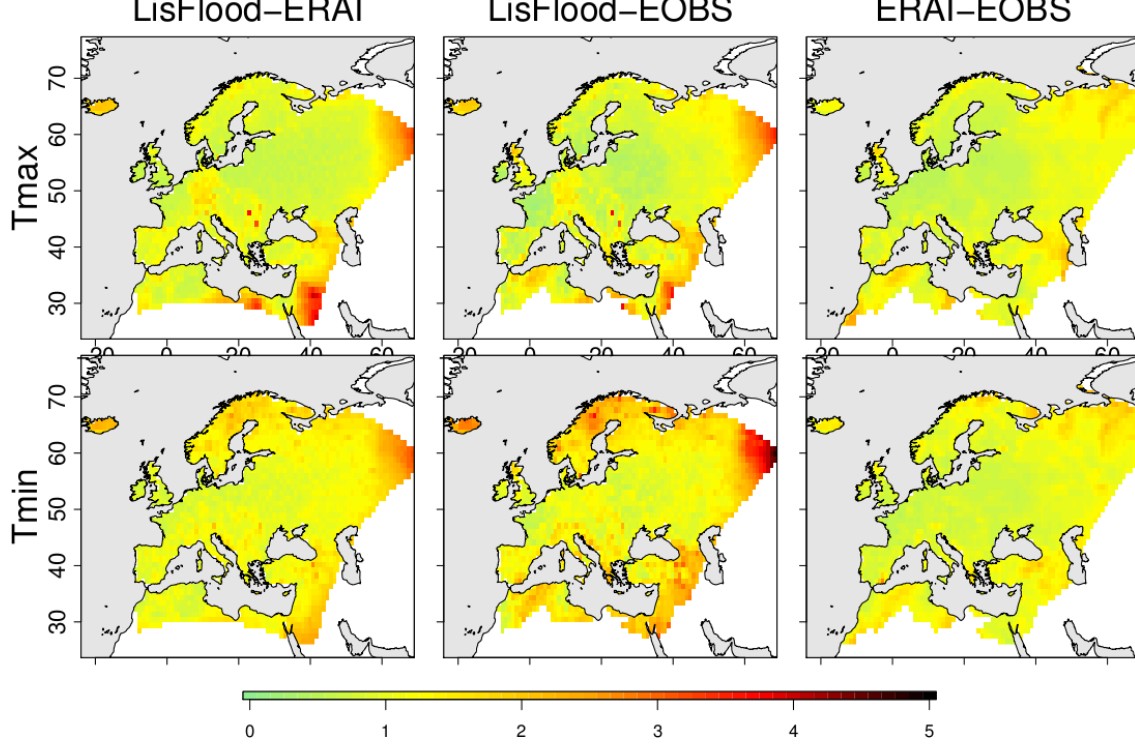


Figure 3 Mean Absolute Error of temperature (in K) between the three datasets, calculated from 1995 to
2015 for Tmin (first row) and Tmax (second row). The datasets compared are indicated on the top of each
column.




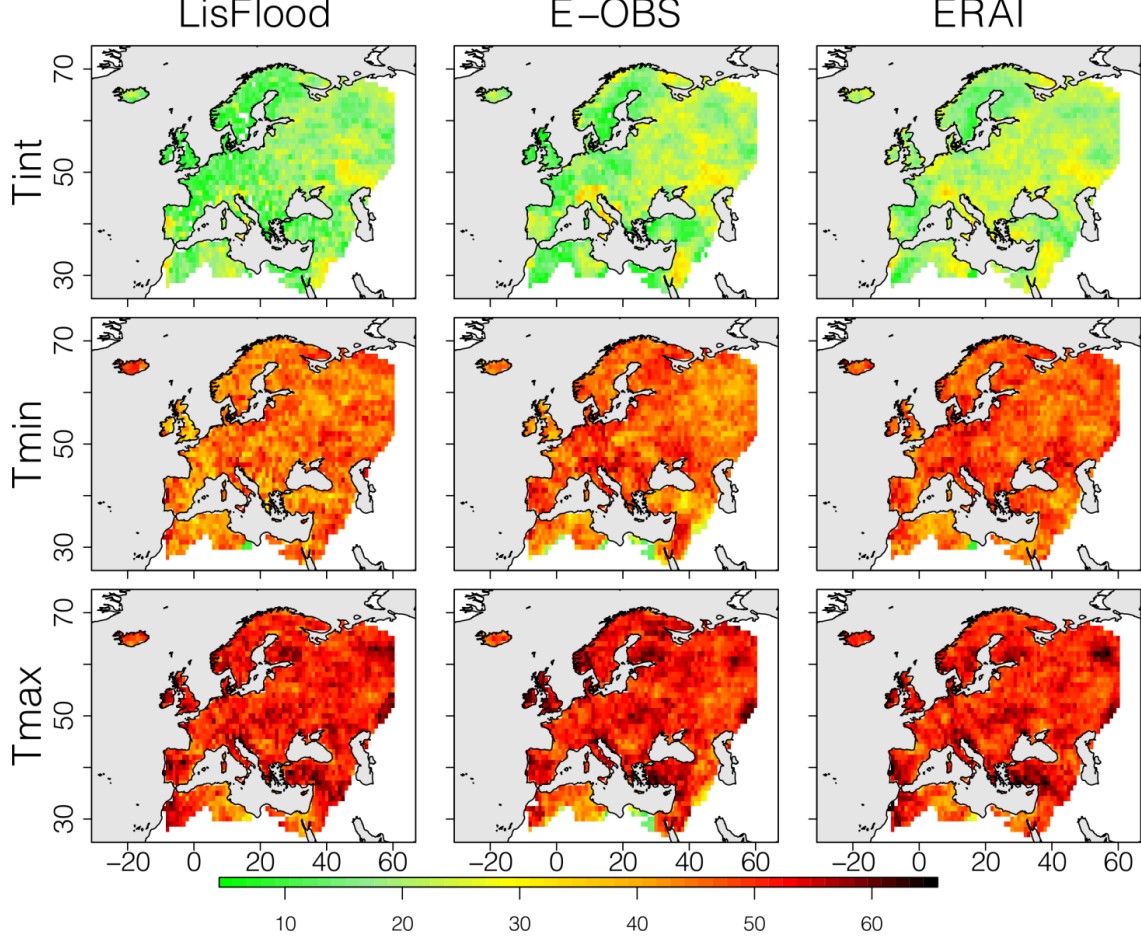


Figure 4 Number of occurrences of heat waves in Europe from 1995 to 2015 using the intersection of both
Tmin and Tmax (Tint, first row), only Tmin (second row), and only Tmax (third row) with LisFlood (first
column), E-OBS (second column) and ERAI (third column) datasets.



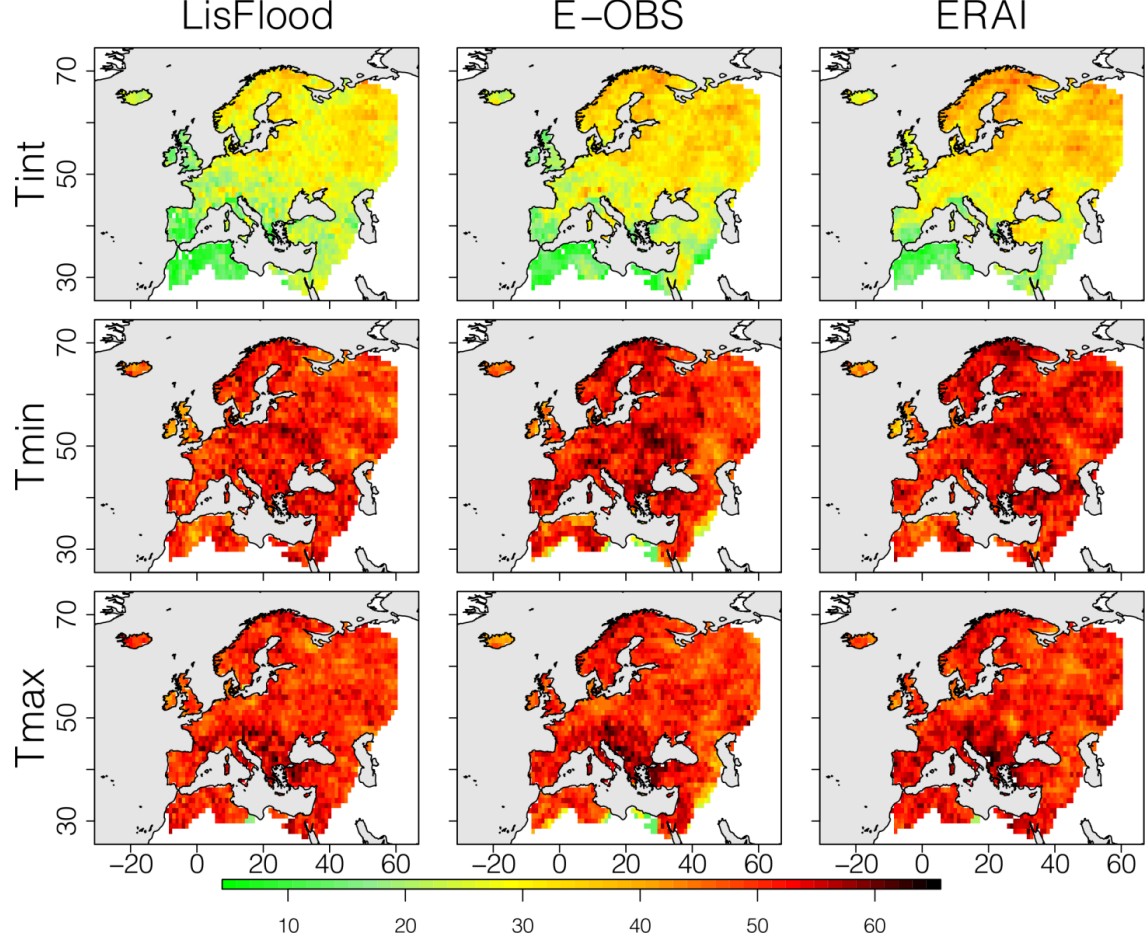


Figure 5 Number of occurrences of cold waves in Europe from 1995 to 2015 using the intersection of both
Tmin and Tmax (Tint, first row), only Tmin (second row), and only Tmax (third row) with LisFlood (first
column), E-OBS (second column) and ERAI (third column) datasets.






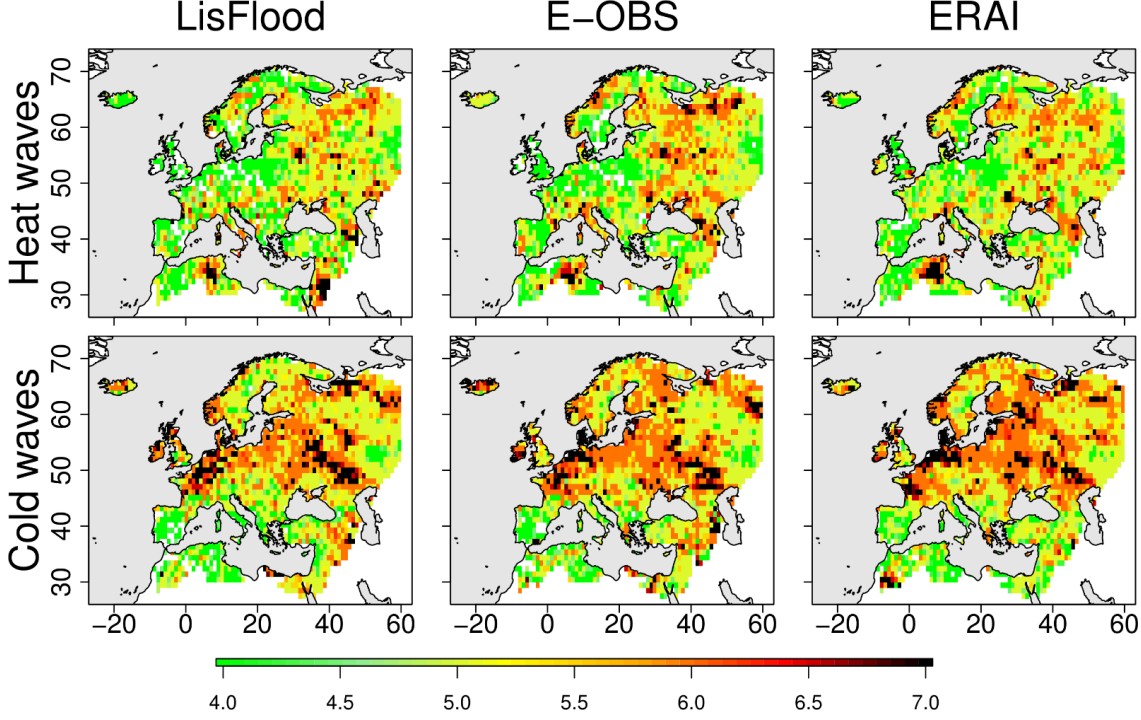


Figure 6 Last quartile of the wave durations (in days) for the heat (top panels) and cold (bottom panels) waves
using LisFlood, E-OBS and ERAI datasets.








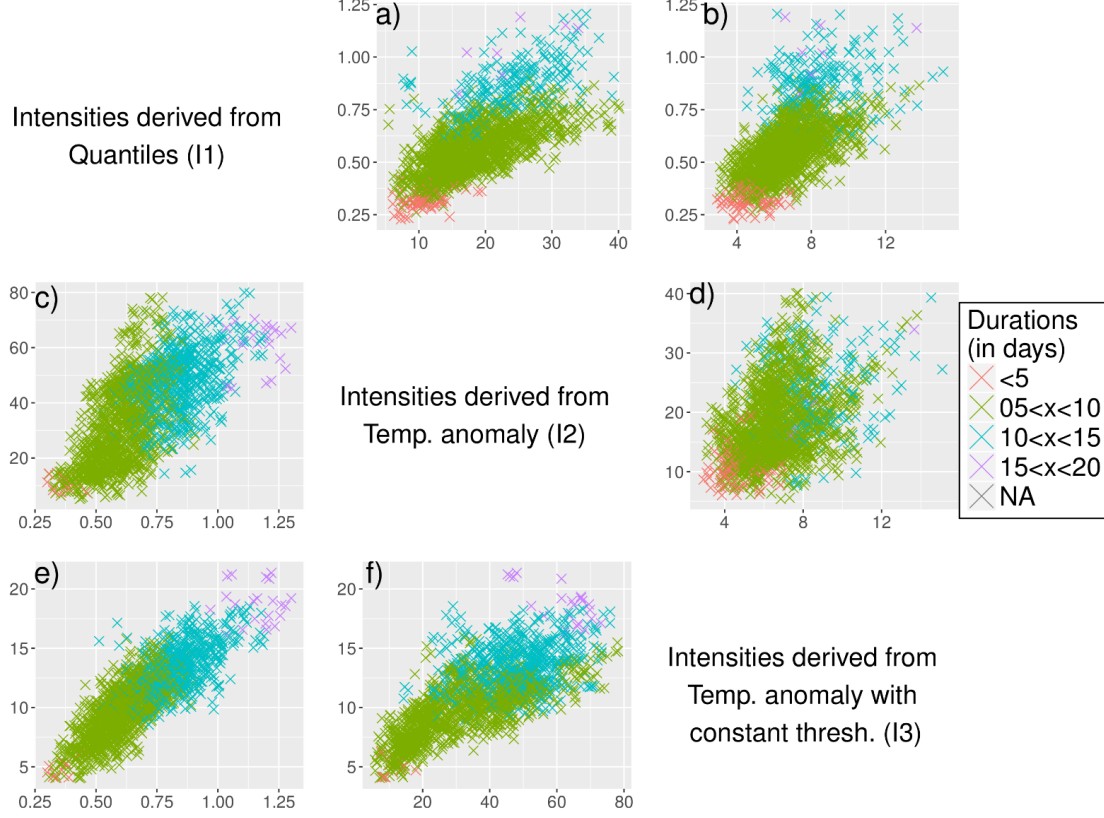


Figure 7 Matrix of scatter plots of the three intensity calculations related to quantiles, temperature anomalies
and temperatures anomalies with constant thresholds (I1, I2 and I3 respectively) during heat (a, b, d) and cold
(c, e, f) waves using LisFlood. The colours indicate the duration (in days) of each wave.





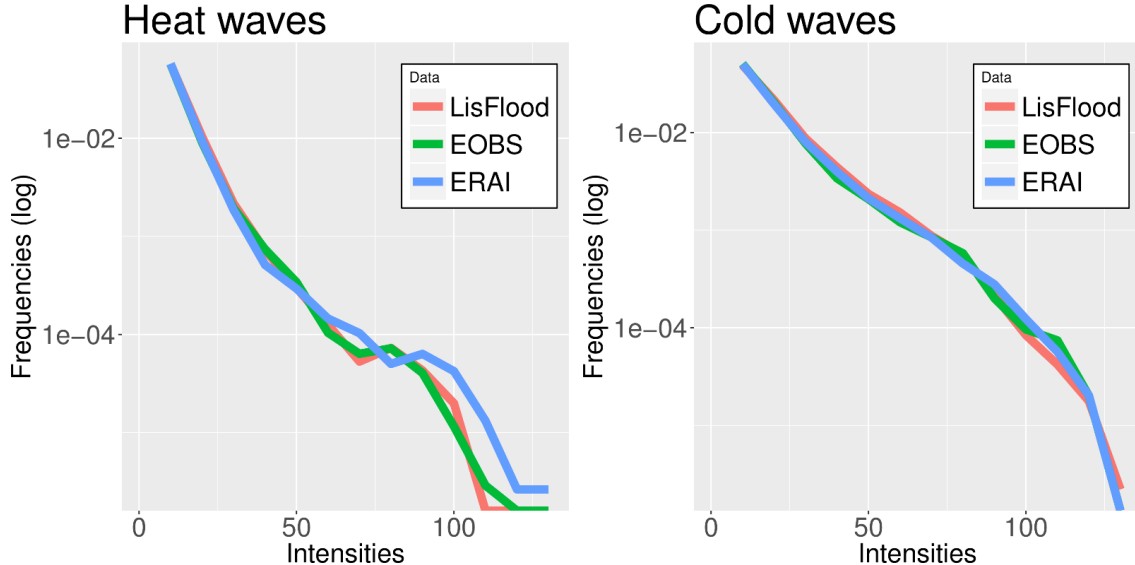


Figure 8 Histograms of heat (left panel) and cold (right panel) waves intensities defined as temperature

anomalies (I2) for the three datasets. Note that the frequency axis are on a Log-scales.





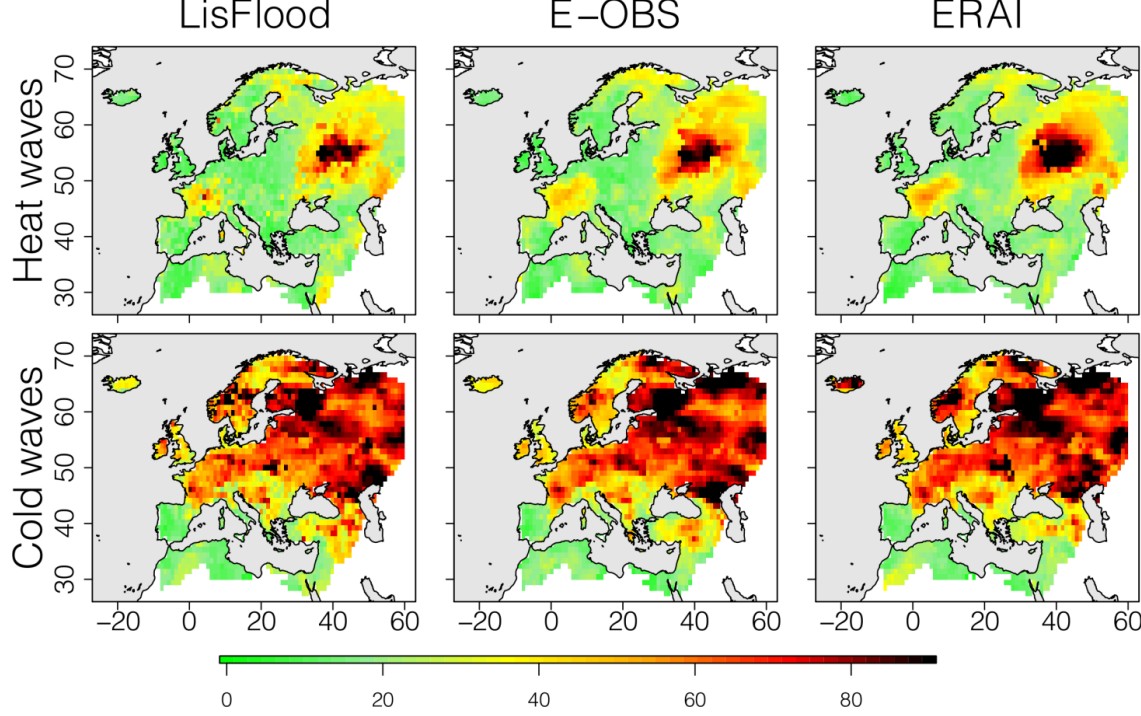


Figure 9 Spatial distribution of the strongest heat (top panels) and cold (bottom panels) waves intensities,
defined as temperature anomalies (I2), using LisFlood, E-OBS and ERAI datasets.




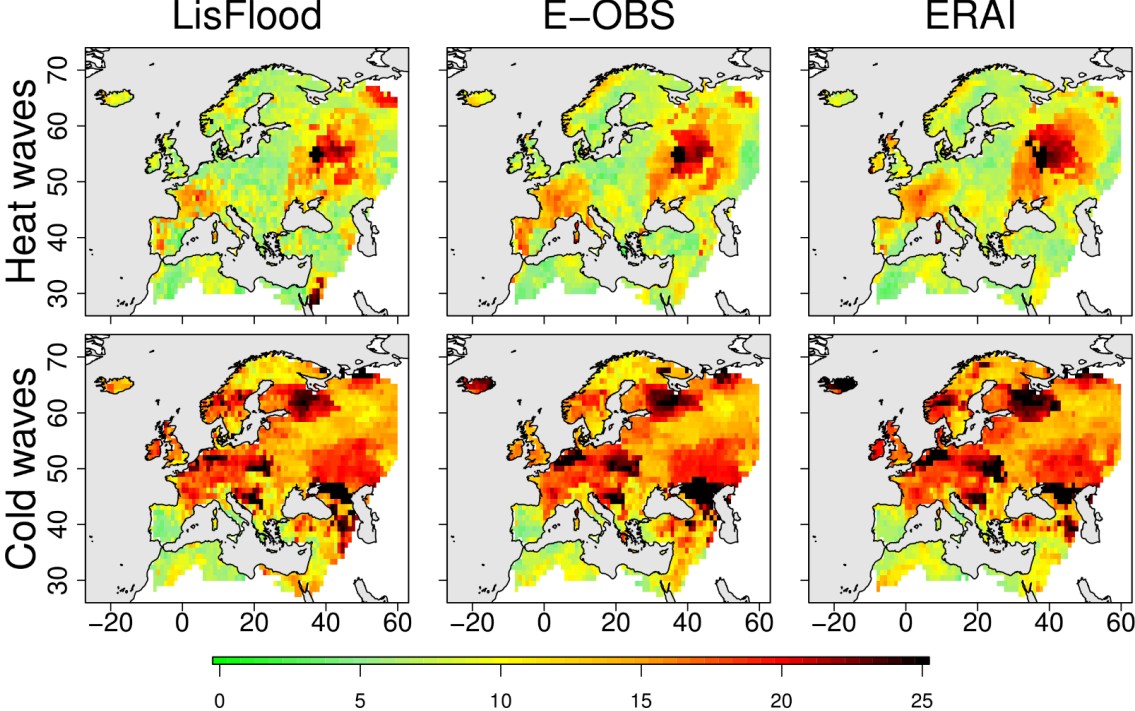


Figure 10 Same as Fig. 9 using the intensity based on the constant threshold (I3) for heat (top panels) and

cold (bottom panels) waves, and based on LisFlood (first row), E-OBS (second row) and ERAI (third row).





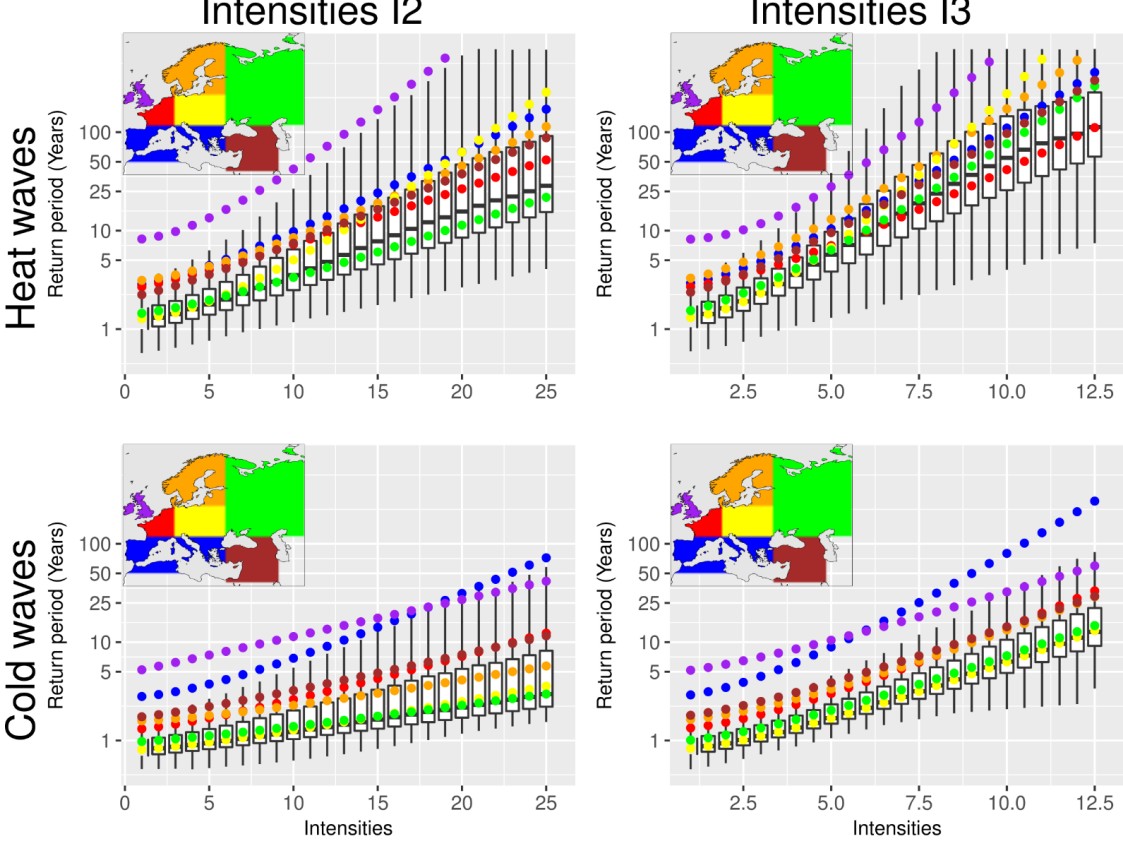


Figure 11 Return periods of monthly intensities of heat (top) and cold (bottom panels) waves for two intensities

(I2, left panels and I3, right panels). Boxes assess the spatial variability for the grid points. Coloured dots
indicate the return period calculated over the regions defined in the small panels.



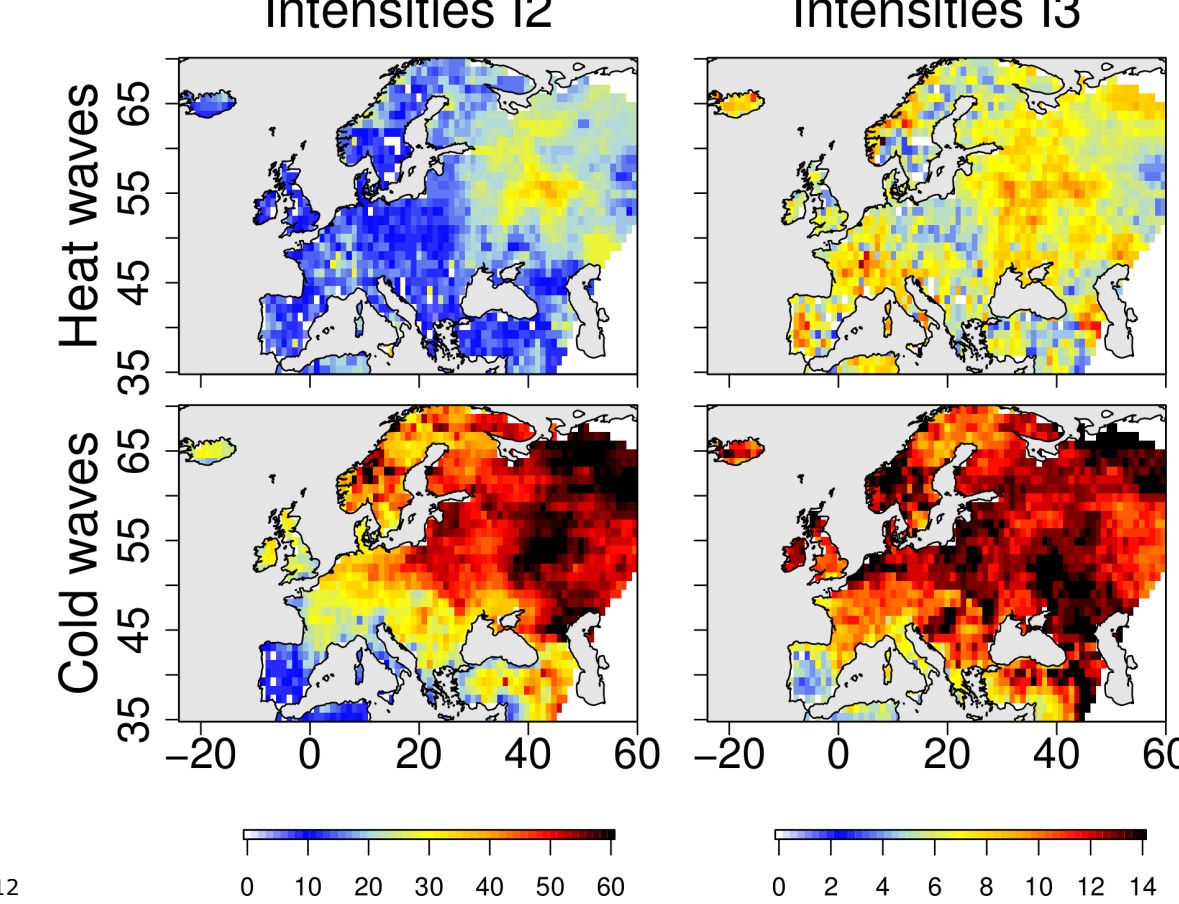


Figure 12 Intensity of the heat (top panels) and cold (bottom panels) waves defined with the temperature
anomalies (I2, left panels), or with constant thresholds (I3, right panels) with a 10-year return period using
LisFlood dataset.