# Peer review of "Towards a monitoring system of temperature extremes in Europe"

_Natural Hazards and Earth System Sciences, 2017_

## Referee Comment (RC1) · Anonymous Referee #1 · 12 Jul 2017

**NHESS-2017-181**

**Towards a monitoring system of temperature extremes in Europe**

Christophe Lavaysse, Carmelo Cammalleri, Alessandro Dosio, Gerard van der Schrier, Andrea Toreti and Jürgen Vogt

General comments:
This is an interesting work, with an interdisciplinary (climate vs health sciences) view, including a reach bibliography in this sense, that fits very well in NHESS and can serve as a reference in discussing concepts like the definition of a heat and cold wave. Nevertheless, I would ask for some clarifications regarding concepts and also for several corrections before final publications. The authors recognise the need of including a reference to human health effects in the definition and intensity determination of heat and cold waves, following WMO recommendations and previous authors statements. In this sense they indicate that the effects on human health can be more related with absolute thermal extremes than to climatological anomalies, but all the work they have done is regarding climatological anomalies, not absolute extremes. The authors are mentioning that the third index of intensity they propose (I3) is oriented to absolute extremes, but I think this is not totally true: the seasonal component of the anomaly (that is clearly influencing both the detection and the I2 intensity) is removed with I3, but no the effect of the geographical, climatic or latitudinal conditioning is not. The reference to calculate differences is always place depending; it depends of the grid point climate, even when the seasonal average is considered to define the reference. I3, for instance, is more measuring the climatological rarity than a risk for the human health, unless the risk for human health is as much depending of the climatological rarity than of the absolute extreme values. In other words, is a summer temperature of 30 degrees in Lapland as much dangerous for human health as 45 degrees in southern Iberia? More clarity about this question would be appropriate in this work, which area of study is geographically/climatologically quite large.

Specific comments/corrections:
-   (Lines 110-113) Are six month periods short enough to be considered as summer/winter events?
-   (Lines 117-118) Is the 11 days' window only used in constructing the calendar of maximum and minimum temperatures?
-   (Lines 135-136) It seems that the only pool accepted within a heat/cold wave is a one-day pool, then the word "less" is not appropriate
-   (Lines 145-150 and tables 1 and 2) The numbers in the tables seem to be space mean/average values. Are they total or time mean/average values (per year?), referred to the 21 years' period? 17 heat waves and 25 cold waves per year, in every grid point, on spatial and time average, seem to be many waves. Around one per year, only, seems to be a small number. The number of hot and cold days' forces to think in the second interpretation. In addition, regarding hot and cold days we can hope equal number of

days within the extremes percentiles (0.9 and 0.1, respectively) for Tmin and Tmax. This is the case for cold days, but not for hot days: why not?

- (Line 156) Are 0.9 and 0.1 the only possible values for Thres?
- (Lines 156, 164, 172) Des N include the pool day, if any?
- (Line 235) I don't understand the first sentence
- (Lines 241-243) There is here a coherent explanation for an experimental result, but it is a quite surprising assessment for me: it is supposed that the oceans are stabilising factors for the temperature, reducing the variability (?)
- (Lines 271-286 and figures 9 and 10) The problem with the 2003 summer in south-western Europe was the repetition of hoy events in the same summer, perhaps not all of them strictly fitting the heat wave definition, but with a dangerous accumulative effect. When taking only one of the events (the strongest heat wave) appears France with a role that perhaps is not very realistic. Perhaps the sum of intensities in the year of larger sum could be a complement or a substitution for the results shown … (?)
- (Table 3) I don't understand this table well. Would you like to be more clear?
- (Figure 7) It seems to me that the squares c) and d) are exchanged

---

## Author Comment (AC1) · 28 Jul 2017

Towards a monitoring system of temperature extremes in Europe Christophe Lavaysse, Carmelo Cammalleri, Alessandro Dosio, Gerard van der Schrier, Andrea Toreti and Jürgen Vogt

**Review 1:** General comments: This is an interesting work, with an interdisciplinary (climate vs health sciences) view, including a reach bibliography in this sense, that fits very well in NHESS and can serve as a reference in discussing concepts like the definition of a heat and cold wave. Nevertheless, I would ask for some clarifications regarding concepts and also for several corrections before final publications. The authors recognise the need of including a reference to human health effects in the definition and intensity

determination of heat and cold waves, following WMO recommendations and previous authors statements. In this sense they indicate that the effects on human health can be more related with absolute thermal extremes than to climatological anomalies, but all the work they have done is regarding climatological anomalies, not absolute extremes. The authors are mentioning that the third index of intensity they propose (I3) is oriented to absolute extremes, but I think this is not totally true: the seasonal component of the anomaly (that is clearly influencing both the detection and the I2 intensity) is removed with I3, but no the effect of the geographical, climatic or latitudinal conditioning is not. The reference to calculate differences is always place depending; it depends of the grid point climate, even when the seasonal average is considered to define the reference. I3, for instance, is more measuring the climatological rarity than a risk for the human health, unless the risk for human health is as much depending of the climatological rarity than of the absolute extreme values. In other words, is a summer temperature of 30 degrees in Lapland as much dangerous for human health as 45 degrees in southern Iberia? More clarity about this question would be appropriate in this work, which area of study is geographically/climatologically quite large.

First, we would like to thank the anonymous reviewer for the complete and fruitful review.

We partially agree with the first comment: That is true the intensity (I3) is still dependent on the climatology and therefore also on the geographical location. The only way to avoid this would be to apply a constant threshold such as 35 or 40 degrees for heat waves and -10 or -20 degrees for cold waves across the entire continent. These definitions are relatively easy to understand due to the potential impacts linked to such extreme temperatures. Nevertheless, this choice could also be criticised. For example, the heat wave in France in 2003 was associated with absolute temperatures close to 40 degrees. These temperatures are, however, regularly overpassed in Spain (46.8 degrees this year in Cordoba) and are relatively closer to the climatology for Southern

Spain, so the impacts are expected to be different for the same absolute temperature. In other words, the impact of heat or cold waves depends also on the societal adaptation, on the available infrastructures (e.g. air conditioning, building insulations), and so on the climatology. For these reasons it is difficult to define the most robust indicator (if it exists at all).

In this study we have, therefore, chosen indicators based on the rarity of the events. The assumption behind this choice is that the lack of adaptation to these rare events will increase their impacts and so this rarity is representing a potential risk. This point will be discussed in the new version of the paper.

Specific comments/corrections: -(Lines 110-113) Are six month periods short enough to be considered as summer/winter events?

The heat and cold waves are calculated during the entire year. However, information on heat waves (cold waves) is provided only in the summer (winter) months as defined. We have chosen to decompose the year into two extended seasons to simplify the message. Nevertheless, the calculation of the heat (cold) waves for the human health impacts (I3) takes into account the seasonal peak (bottom respectively) of temperature and so favours the waves occurring during the core of the season.

-(Lines 117-118) Is the 11 days' window only used in constructing the calendar of maximum and minimum temperatures?

As mentioned in the text, the 11-day windows are only used to increase the sampling when the CDF of temperature is generated for the quantile calculation.

-(Lines 135-136) It seems that the only pool accepted within a heat/cold wave is a one-day pool, then the word "less" is not appropriate.

Modified as suggested.

-(Lines 145-150 and tables 1 and 2) The numbers in the tables seem to be space mean/average values. Are they total or time mean/average values (per year?), referred

to the 21 years'period? 17 heat waves and 25 cold waves per year, in every grid point, on spatial and time average, seem to be many waves. Around one per year, only, seems to be a small number. The number of hot and cold days' forces to think in the second interpretation. In addition, regarding hot and cold days we can hope equal number of days within the extremes percentiles (0.9 and 0.1, respectively) for Tmin and Tmax. This is the case for cold days, but not for hot days: why not?

The captions of the tables are now clarified. Here, the spatial mean occurrence of waves during the entire period is plotted (i.e. the second interpretation of the reviewer is correct). The number could be interpreted as small, but the objective of the study is to focus on the most extreme events that are, by definition, limited. Also there is a spatial variability of the frequency of occurrence as shown Fig. 4 or Fig. 5, and these occurrences are sometimes larger than 30 (40) for the heat (cold) waves.
The reviewer is right about the constant hot and cold days using only Tmin and Tmax. Nevertheless, two effects will influence the results:

- the presence of undefined values that modify the sampling size

- temperature (quantile) values equal to the thresholds

That occurs both in summer and winter. In winter, by 'chance' the samples are equal to the first percentile of the total population; nevertheless, the standard deviations, which are different to zero, indicate the presence of the same artefact in all cases.

-(Line 156) Are 0.9 and 0.1 the only possible values for Thres?

As discussed in the introduction section, these values are the most common used in the literature and allow to detect enough extreme events to perform statistics. In a previous version of this work other thresholds were chosen. The number of events are, of course, strongly influenced. But most of the spatial variabilities and the coherence between the different datasets are retained.

-(Lines 156, 164, 172) Does N include the pool day, if any?

No, as discussed in the chapter (l136-137), the pool day is not taken into account in the calculation of the intensity. This sentence has been clarified.

-(Line 235) I don't understand the first sentence

The sentence was rephrased. The meaning of this sentence is to introduce the mean duration of the waves.

-(Lines 241-243) There is here a coherent explanation for an experimental result, but it is a quite surprising assessment for me: it is supposed that the oceans are stabilising factors for the temperature, reducing the variability (?)

That is a good point. Our hypothesis about the increase of duration close to the Baltic Sea is based on the the oceans' stabilizing effect on the temperature and therefore a signal with lower high frequency modulations. In this case, if an anomaly occurs, it has a bigger chance to be longer and so potentially creates longer heat/cold waves. This is due to our detection method of HW and CW that is based on the quantiles and not on absolute temperature. The latter is generally less variable and less extreme along the coasts. To illustrate this behavior, the wavelet analysis (Torence and Compo, 1998) of temperature in winter and summer is calculated. To focus on the high frequency variabilities, the power of the Lorentz boxes with periodicity lower than 3 days are summed up. Regions with large (low) high frequency modulations are indicated with large (low) power. This method is quite close to a classical FFT, but it is more powerful with information in both time and frequency. In the Fig. 1, the regions with low modulations (Eastern Europe in summer or Northern Russia and north of Poland in winter) are also the regions with high frequency of occurrence or with longer durations.

-(Lines 271-286 and figures 9 and 10) The problem with the 2003 summer in south-western Europe was the repetition of hot events in the same summer, perhaps not all of them strictly fitting the heat wave definition, but with a dangerous accumulative effect.

When taking only one of the events (the strongest heat wave) appears France with a role that perhaps is not very realistic. Perhaps the sum of intensities in the year of larger sum could be a complement or a substitution for the results shown ...(?)

As explained in the paper, the operational system will sum up all the intensities of heat/cold waves that occurred during the last month, the last couple of months and the entire season. The purpose is exactly to fulfil the characteristic mentioned by the reviewer to take into account the repetition of several events in a row. This will be implemented operationally and the time span considered depends on the month. Table 3 indicates for each month which indicator will be calculated.

-(Table 3) I don't understand this table well. Would you like to be more clear?

Please see the previous comment. The caption of the table has been changed to clarify it.

-(Figure 7) It seems to me that the squares c) and d) are exchanged

No, square c is for the comparison in between I1 and I2 in winter.
* * *
[Figure]

[Figure]

**Fig. 1.** Sum of the power of Lorenz boxes (with periodicity lower than 3 days) of the wavelet analysis of the merged Tmin and Tmax, in summer (left) and in winter (right).

---

## Referee Comment (RC2) · Anonymous Referee #2 · 30 Aug 2017

This is an interesting study that quantify the intensity of heat and cold waves regarding the climatology for the development of a monitoring system of temperature extremes in Europe. This study represent a substantial contribution to the understanding of natural hazards and their consequences. Explanations, results and references are appropriate, and are presented in a clear, concise and well-structured way. Figures and tables are helpful and the number and quality of both is appropriated. Overall, I found it is an interesting paper and would recommend publication after some additional work. Although I enjoyed reading the manuscript, the paper is written well and I appreciate the work of the authors, I have some concerns about the methodological choices.

My main concern is related to the small size of the sample and the return times computation. Although climate is usually defined as an average of weather, the classical

period as defined by the World Meteorological Organization (WMO) is 30 years. So the most important caveat that I see in this study is the small sample of 21 years. The return times are computed with the intensities of the waves you have detected with a climatology of 21 years, It is also difficult to believe in return times greater than 100 years (figure 11) computed over a basis of 21 years. I also understand that Lisflood has more benefits than the other two for the monitoring system but it is quite short dataset (starting at 1990).

Therefore, and in order to validate and justified the short period of study, I suggest to repeat the experiment (including some additional figures or tables in the manuscript) but using just EOBs for the whole period (1950-2015) since is the largest dataset you have used and has a good agreement with Lisflood. Hence, you will have a largest climatology to detect the waves and compute the return times with less influence of noise due to the small sample size. If these results are in consistence with the ones you got using the 21 common years of the three datasets your results and the monitoring system, which, by the way, I find very interesting and promising, will demonstrate that are robust enough even for a short period, and so the use of Lisflood will be justified for this purpose.

---

## Author Comment (AC2) · 10 Oct 2017

**This is an interesting study that quantify the intensity of heat and cold waves regarding the climatology for the development of a monitoring system of temperature extremes in Europe. This study represent a substantial contribution to the understanding of natural hazards and their consequences. Explanations, results and references are appropriate, and are presented in a clear, concise and well-structured way. Figures and tables are helpful and the number and quality of both is appropriated. Overall, I found it is an interesting paper and would recommend publication after some additional work. Although I enjoyed reading the manuscript, the paper is written well and I appreciate the work of the authors, I have some concerns about the methodological choices.**

[Figure]

We would like to thank the reviewer for his fruitful and positive comments. Please find below the responses of the comments. The article has been revised according to the suggestions of all the reviewers.

**My main concern is related to the small size of the sample and the return times computation. Although climate is usually defined as an average of weather, the classical period as defined by the World Meteorological Organization (WMO) is 30 years. So the most important caveat that I see in this study is the small sample of 21 years. The return times are computed with the intensities of the waves you have detected with a climatology of 21 years, it is also difficult to believe in return times greater than 100 years (figure 11) computed over a basis of 21 years. I also understand that Lisflood has more benefits than the other two for the monitoring system but it is quite short dataset (starting at 1990).**
**Therefore, and in order to validate and justified the short period of study, I suggest to repeat the experiment (including some additional figures or tables in the manuscript) but using just EOBs for the whole period (1950-2015) since is the largest dataset you have used and has a good agreement with Lisflood. Hence, you will have a largest climatology to detect the waves and compute the return times with less influence of noise due to the small sample size. If these results are in consistence with the ones you got using the 21 common years of the three datasets your results and the monitoring system, which, by the way, I find very interesting and promising, will demonstrate that are robust enough even for a short period, and so the use of Lisflood will be justified for this purpose. This is a very interestcomment. The size of the samples and the extrapolation of the data are always an important and sensitive point.**

First of all, the recommendation of WMO could be slightly different depending the purpose of the study: climate evolution, detection of extreme, climatological reference,

climatological evolution of extremes etc... However, there is no clear consensus according to WMO (2009) about a specific duration. As the purpose of this monitoring system is not to assess the climatological trend of the extreme events, as done by Russo et al. (2015), but it is on the detection of relative intense events according to a reference period we believe that a shorter time series (20 years instead of 30) is sufficient. This baseline duration is not considered as too short and plenty of study/datasets are using this duration period (Kharin et al. 2013, Vautard et al. 2013, Monhart et al. 2016). It is also possible to mention that ECMWF runs an extended ensemble model twice a week that going up to 45-day lead time (Vitart, 2004). In parallel, the European Centre runs hindcast (or reforecast) to create a climatological baseline (to correct bias of the model, built climatology and detect the strongest anomalies). These hindcasts are also performed using 20 years highlighting the usefulness of this length of climatological reference. According to the WMO guideline and the mentioned previous studies, but also due to two technical reasons i) the availability of the datasets and ii) to be consistent with the forecasts that will be implemented in the same system in the future, we decide to keep the 20-year climatology to detect and characterize the intensities of heat and cold waves.

Moreover, the suggestion of the reviewer about using the full 50-year period to perform the climatology and the return period is interesting but questionable. Indeed, it is well known that Europe endures a significant evolution of the climate and of the extreme temperatures generating a non-stationary occurrence/intensity of heat and cold waves (Gonzales-Hidalgo et al. 2016). This is especially true if we consider a long period (such as 50-year). In that non-stationary context, using a 'too' long sampling to compute the return period of extreme events will generate an underestimation of these values comparing to the more recent climatology. For this reason we consider non relevant to perform this long-term reference that has been already study for other purposes (Russo et al., 2016). To better identify and extrapolate the return periods of rare events, the fitting of the observed extreme event onto a parametric distribution is

a robust and common method employed in the literature (Coles et al. 2001, Schar et al. 2004, Blender et al. 2008). According to significant tests employed in that study to guarantee the robustness of the distribution, it exists uncertainties for return periods larger than the duration of the observed sampling. For these reasons, and according to the reviewer's comment, in Figure 11 of the revised manuscript, return periods longer than 25-years are indicated with grey shadows and, in addition, the x-axis is reduced in order to have less than 50

Finally about the use of the Lisflood datasets, according to the WMO report previously mentioned, we believe that a 20 year period is long enough to provide robust climatology of the events for this purpose; furthermore, this database is also the most accurate and dense observed datasets of temperature we can access operationally with a very short delay (about 24h). For these consideration, we are still considering this dataset as the most suitable for our purposes. Considering our study, we have also verified that the results and the climatology obtained using the dataset used in this study is very close to other well-known and commonly used datasets (E-OBS and ERAI reanalysis) although they are released too late for use in an operational system as the one proposed in our work. In the future, depending the possibly to get new datasets from Copernicus CCS, we could change the datasets.

Bibliography:

Blender, Richard, K. Fraedrich, and Frank Sienz. "Extreme event return times in long-term memory processes near 1/f." Nonlinear Processes in Geophysics 15.4 (2008): 557.

Coles, Stuart, et al. An introduction to statistical modeling of extreme values. Vol. 208. London: Springer, 2001. Gonzalez‐Hidalgo, José Carlos, et al. "Recent trend in temperature evolution in Spanish mainland (1951–2010): from warming to hiatus."

International Journal of Climatology 36.6 (2016): 2405-2416

Kharin, Viatcheslav V., et al. "Changes in temperature and precipitation extremes in the CMIP5 ensemble." Climatic change 119.2 (2013): 345-357.

Monhart, Samuel, et al. "Verification of ECMWF monthly forecasts for the use in hydrological predictions." EGU General Assembly Conference Abstracts. Vol. 18. 2016.

Russo, Simone et al. "Top ten European heatwaves since 1950 and their occurrence in the coming decades." Environmental Research Letters, 2015, 10.12: 124003.

Russo, Simone, et al. "When will unusual heat waves become normal in a warming Africa?" Environmental Research Letters, 2016, 11.5: 054016.

Schar, Christoph, et al. "The role of increasing temperature variability in European summer heatwaves." Nature 427.6972 (2004): 332.

Vautard, Robert, et al. "The simulation of European heat waves from an ensemble of regional climate models within the EURO-CORDEX project." Climate dynamics 41.9-10 (2013): 2555-2575.

Vitart, Frédéric. "Monthly forecasting at ECMWF." Monthly Weather Review 132.12 (2004): 2761-2779. DATA, Climate. Guidelines on analysis of extremes in a changing climate in support of informed decisions for adaptation. World Meteorological Organization, 2009.